# Learning from Label Proportions with Generative Adversarial Networks

**Jiabin Liu**[*]
Samsung Research China - Beijing
Beijing 100028, China
liujiabin008@126.com

**Bo Wang**[*]
University of International Business and Economics
Beijing 100029, China
wangbo@uibe.edu.cn

**Zhiquan Qi**[†]  **Yingjie Tian**  **Yong Shi**
University of Chinese Academy of Sciences
Beijing 100190, China
qizhiquan@foxmail.com, {tyj,yshi}@ucas.ac.cn

## Abstract

In this paper, we leverage generative adversarial networks (GANs) to derive an effective algorithm LLP-GAN for learning from label proportions (LLP), where only the bag-level proportional information in labels is available. Endowed with end-to-end structure, LLP-GAN performs approximation in the light of an adversarial learning mechanism, without imposing restricted assumptions on distribution. Accordingly, we can directly induce the final instance-level classifier upon the discriminator. Under mild assumptions, we give the explicit generative representation and prove the global optimality for LLP-GAN. Additionally, compared with existing methods, our work empowers LLP solver with capable scalability inheriting from deep models. Several experiments on benchmark datasets demonstrate vivid advantages of the proposed approach.

## 1 Introduction

Deep learning benefits from *end-to-end* design philosophy, which emphasizes minimal a priori representational and computational assumption, and greatly avoids explicit structure and "hand-engineering" [4]. Doubtless, most of its achievements are affirmed by the access to the abundance of fully supervised data [14, 13, 26]. One reason is that a large amount of complete data can alleviate the over-fitting problem without deteriorating the hypothesis set complexity (e.g., a sophisticated network design with vast parameters).

Unfortunately, fully labeled data is not always handy to utilize. Firstly, it is infeasible or labor-intensive to obtain abundant accurate labeled data [31]. Secondly, labels are not accessible under certain circumstances, such as privacy constraints [22]. Therefore, the community begins to pay attention to *weakly supervised learning* (a.k.a., *weakly labeled learning*, WeLL), concerning any corrosion in supervised information. For example, semi-supervised learning (SSL) is a WeLL problem by concealing most of the labels in the training stage. Another widespread WeLL problem is multi-instance learning (MIL) [19], which is also a representative in *learning with bags*.

Furthermore, generative adversarial networks (GANs) [11], which is originally proposed to synthesize high fidelity data in line with the estimation of underlying data distribution, can be potentially applied to WeLL. For instance, GANs can deal with $K$-class SSL [28], by treating the generated data as

---

[*]Joint first authors with equal contribution
[†]Corresponding author

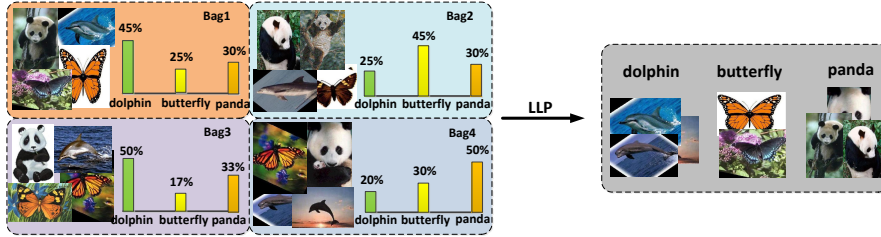

Figure 1: An illustration of multi-class learning from label proportions. In detail, the data belongs to three categories and is partitioned into four non-overlapping groups. In each group, the sizes of green, blue, and orange rectangles respectively denote available label proportions in different categories. We only know the sample feature information and class proportions in every group.

class $K+1$ and exploiting feature matching (FM) as the generator objective. Unlike performing maximizing log-likelihood on the variational lower bound of unlabeled data [16, 17], GANs seeks the equilibrium between two networks (discriminator and generator) by alternatively upgrading in an adversarial game, and directly obtains the final classifier upon the discriminator.

In this paper, we push the envelope further by focusing on applying GANs to another WeLL problem: learning from label proportions (LLP) (see [23, 27, 33] for real-life applications). We illustrate multi-class LLP problem in Figure 1. By referring *group* as *bag*, LLP also fits for *learning with bags* settings, which is primarily established in MIL [9]. In LLP, we strive for an instance-level multi-class classifier merely with *multi-bag* proportional information and instance features (inputs). On the right, instances from different categories are classified based on a well-trained multi-class classifier.

The main challenge for LLP is to shrink the uncertainty in label inference based on the bag-level proportional information. Before deep learning making its appearance, several shallow methods have been proposed, such as probability estimation methods (e.g., MeanMap [23] and Laplacian MeanMap [21]) and SVM-based methods (e.g., InvCal [27] and alter-$\propto$SVM [34, 22]). However, statistical approaches are extremely constrained by strict assumption on data distribution and prior knowledge, while the SVM-based methods suffer from the NP-hard combinatorial optimization issue, thus is lack of scalability.

The motivation of our work mainly lies in the following three aspects. Firstly, as introduced above, GAN is an elegant recipe for solving WeLL problems, especially SSL [28]. From this viewpoint, our approach is in line with the idea of applying GAN to incomplete label scenarios. More importantly, the success of generative models for WeLL stems from the explicit or implicit representation learning, which has been an essential method for unsupervised learning for a long time [5, 24], e.g., VAE [16]. In our approach, the convolution layers in discriminator can perform as a feature extractor for downstream tasks, which is proved to be efficient [24]. Hence, our work can be regarded as *solving LLP based on representation learning with GANs*. In this scheme, generated fake samples encourage the discriminator to not only detect the difference between the real and the fake instances, but also distinguish *true $K$ classes* for real samples (through $K+1$ classifier). Thirdly, most LLP methods assume that the bags are i.i.d. [23, 34], which cannot sufficiently explore the underlying distribution in the data and may be contradicted in certain applications. Instead, the generator in LLP-GAN is designated to learn data distributions through the adversarial scheme without this assumption.

The remainder of this paper is organized as follows:

- In Section 2, we give preliminaries regarding LLP problem and propose a simple improvement based on entropy regularization for the existing deep LLP solver.

- In Section 3, we describe our adversarial learning framework for LLP, especially the lower bound of discriminator. In particular, we reveal the relationship between prior class proportions and posterior class likelihoods. More importantly, we offer a decomposition representation of the class likelihood with respect to the prior class proportions, which verifies the existence of the final classifier.

- In Section 4, we empirically show that our method can achieve SOTA performance on large-scale LLP problems with a low computational complexity.

## 2 Preliminaries

This section offers necessary preliminaries for our approach, including the formal problem setting and related work with simple extensions.

### 2.1 The Multi-class LLP

Before further discussion, we formally describe multi-class LLP. For simplicity, we assume that all the bags are disjoint and let $\mathcal{B}_i = \{\mathbf{x}_i^1, \mathbf{x}_i^2, \cdots, \mathbf{x}_i^{N_i}\}, i = 1, 2, \cdots, n$ be the bags in training set. Then, training data is $\mathcal{D} = \mathcal{B}_1 \cup \mathcal{B}_2 \cup \cdots \cup \mathcal{B}_n, \mathcal{B}_i \cap \mathcal{B}_j = \emptyset, \forall i \neq j$, where the total number of bags is $n$.

Assuming we have $K$ classes, for $\mathcal{B}_i$, let $\mathbf{p}_i$ be a $K$-element vector, where the $k^{th}$ element $p_i^k$ is the proportion of instances belonging to the class $k$, with the constraint $\sum_{k=1}^K p_i^k = 1$, i.e.,

$$p_i^k := \frac{|\{j \in [1:N_i] | \mathbf{x}_i^j \in \mathcal{B}_i, y_i^{j*} = k\}|}{|\mathcal{B}_i|}. \tag{1}$$

Here, $[1:N_i] = \{1, 2, \cdots, N_i\}$ and $y_i^{j*}$ is the unaccessible ground-truth instance-level label of $\mathbf{x}_i^j$. In this way, we can denote the available training data as $\mathcal{L} = \{(\mathcal{B}_i, \mathbf{p}_i)\}_{i=1}^n$. The goal of LLP is to learn an instance-level classifier based on $\mathcal{L}$.

### 2.2 Deep LLP Approach

In terms of deep learning, DLLP firstly leverages DNNs to solve multi-class LLP problem [1]. Using DNN's probabilistic classification outputs, it is straightforward to adapt cross-entropy loss into a bag-level version by averaging the probability outputs in every bag as the proportion estimation. To this end, inspired by [31], DLLP reshapes standard cross-entropy loss by substituting instance-level label with label proportion, in order to meet the requirement of proportion consistency.

In detail, suppose that $\tilde{\mathbf{p}}_i^j = p_\theta(\mathbf{y}|\mathbf{x}_i^j)$ is the vector-valued DNNs output for $\mathbf{x}_i^j$, where $\theta$ is the network parameter. Let $\oplus$ be element-wise summation operator. Then, the bag-level label proportion in the $i^{th}$ bag is obtained by incorporating the element-wise posterior probability:

$$\overline{\mathbf{p}}_i = \frac{1}{N_i} \bigoplus_{j=1}^{N_i} \tilde{\mathbf{p}}_i^j = \frac{1}{N_i} \bigoplus_{j=1}^{N_i} p_\theta(\mathbf{y}|\mathbf{x}_i^j). \tag{2}$$

Different from the discriminant approaches, in order to smooth *max* function [6], $\tilde{\mathbf{p}}_i^j$ is in a vector-type softmax manner to produce the probability distribution for classification. Taking *log* as element-wise logarithmic operator, the objective of DLLP can be intuitively formulated using cross-entropy loss $L_{prop} = -\sum_{i=1}^n \mathbf{p}_i^\intercal log(\overline{\mathbf{p}}_i)$. It penalizes the difference between prior and posterior probabilities in bag-level, and commonly exists in GAN-based SSL [29].

### 2.3 Entropy Regularization for DLLP

Following the entropy regularization strategy [12], we can introduce an extra loss $E_{in}$ with a trade-off hyperparameter $\lambda$ to constrain instance-level output distribution in a low entropy accordingly:

$$L = L_{prop} + \lambda E_{in} = -\sum_{i=1}^n \mathbf{p}_i^\intercal log(\overline{\mathbf{p}}_i) - \lambda \sum_{i=1}^n \sum_{j=1}^{N_i} (\tilde{\mathbf{p}}_i^j)^\intercal log(\tilde{\mathbf{p}}_i^j). \tag{3}$$

This straightforward extension of DLLP is similar to a KL divergence, taking care of bag-level and instance-level consistencies simultaneously. It takes advantage of DNN's output distribution to cater to the label proportions requirement, as well as minimizing output entropy as a regularization term to guarantee high true-fake belief. This is believed to be linked with an inherent maximum a posteriori (MAP) estimation [6] with certain prior distribution in network parameters. However, we will not look at the performance of this extension and consider not to include it as a baseline, because the experimental results empirically suggest that the original DLLP has already converged to the solution with fairly low instance-level entropy, which makes the proposed regularization term redundant. We offer results of this empirical study in the Supplementary Material.

# 3 Adversarial Learning for LLP

In this section, we focus on LLP based on adversarial learning and propose LLP-GAN, which devotes GANs to harnessing LLP problem.

We illustrate the LLP-GAN framework in Figure 2. Firstly, the generator is employed to generate image with input noise, which is labeled as fake, and the discriminator yields class confidence maps for each class (including the fake one) by taking both fake and real data as its inputs. This results in the *adversarial loss*. Secondly, we incorporate the proportions by adding the *cross entropy loss*.

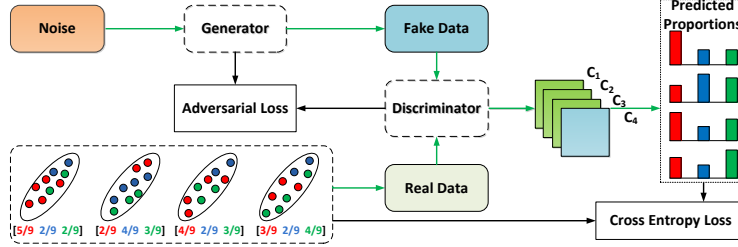

Figure 2: An illustration of our LLP-GAN framework.

## 3.1 The Objective Function of Discriminator

In LLP-GAN, our discriminator is not only to identify whether a sample is from the real data or not, but also to elaborately distinguish each real input's label assignment as a $K$ classes classifier. We incorporate the unsupervised adversarial learning into the $L_{unsup}$ term.

Next, the main issue becomes how to exploit the proportional information to guide this unsupervised learning correctly. To this end, we replace the supervised information in semi-supervised GANs with label proportions, resulting in $L_{sup}$, same as $L_{prop}$ in (3).

**Definition 1.** *Suppose that $\mathcal{P}$ is a partition to divide the data space into $n$ disjoint sections. Let $p_d^i(\mathbf{x}), i = 1, 2, \cdots, n$ be marginal distributions with respect to elements in $\mathcal{P}$ respectively. Accordingly, $n$ bags in LLP training data spring from sampling upon $p_d^i(\mathbf{x}), i = 1, 2, \cdots, n$. In the meantime, let $p(\mathbf{x}, y)$ be the unknown holistic joint distribution.*

We normalize the first $K$ classes in $P_D(\cdot|\mathbf{x})$ into the instance-level posterior probability $\tilde{p}_D(\cdot|\mathbf{x})$ and compute $\overline{\mathbf{p}}$ based on (2). Then, the *ideal* optimization problem for the discriminator of LLP-GAN is:

$$\max_D \ V(G, D) = L_{unsup} + L_{sup} = L_{real} + L_{fake} - \lambda CE_{\mathcal{L}}(\mathbf{p}, \overline{\mathbf{p}})$$

$$= \sum_{i=1}^n E_{\mathbf{x} \sim p_d^i}\Big[log P_D(y \leq K|\mathbf{x})\Big] + E_{\mathbf{x} \sim p_g}\Big[log P_D(K+1|\mathbf{x})\Big] + \lambda \sum_{i=1}^n \mathbf{p}_i^\mathsf{T} log(\overline{\mathbf{p}}_i). \quad (4)$$

Here, $p_g(\mathbf{x})$ is the distribution of the synthesized data.

**Remark 1.** *When $P_D(K+1|\mathbf{x}) \neq 1$, the normalized instance-level posterior probability $\tilde{p}_D(\cdot|\mathbf{x})$ is:*

$$\tilde{p}_D(k|\mathbf{x}) = \frac{P_D(k|\mathbf{x})}{1 - P_D(K+1|\mathbf{x})}, k = 1, 2, \cdots, K. \quad (5)$$

*If $P_D(K+1|\mathbf{x}) = 1$, let $\tilde{p}_D(k|\mathbf{x}) = \frac{1}{K}, k = 1, 2, \cdots, K$. Note that weight $\lambda$ in (4) is added to balance between supervised and unsupervised terms, which is a slight revision of SSL with GANs [28, 8]. Intuitively, we reckon that the proportional information is too weak to fulfill supervised learning pursuit. Hence, a relatively large weight should be preferable in the experiments. However, large $\lambda$ may result in unstable GANs training. For simplicity, we fix $\lambda = 1$ in the following theoretical analysis on discriminator.*

Aside from identifying the first two terms in (4) as that in semi-supervised GANs, the cross-entropy term harnesses the label proportions consistency. In order to justify the non-triviality of this loss, we first look at its lower bound. More importantly, it is easier to perform the gradient method on the lower bound, because it swaps the order of *log* and the summation operation. For brevity, the analysis will be done in a non-parametric setting, i.e., we assume that both $D$ and $G$ have infinite capacity.

**Remark 2** (**The Lower Bound Approximation**). *Let $p_i(k)$ be the class $k$ proportion in the $i^{th}$ bag. According to the idea of sampling methods and Jensen's inequality, we have:*

$$-CE_{\mathcal{L}}(\mathbf{p}, \overline{\mathbf{p}}) = \sum_{i=1}^{n}\sum_{k=1}^{K} p_i(k) log\Big[\frac{1}{N_i}\sum_{j=1}^{N_i} \tilde{p}_D(k|\mathbf{x}_i^j)\Big]$$

$$\simeq \sum_{i=1}^{n}\sum_{k=1}^{K} p_i(k) log\Big[\int p_d^i(\mathbf{x})\tilde{p}_D(k|\mathbf{x})d\mathbf{x}\Big] \geqslant \sum_{i=1}^{n}\sum_{k=1}^{K} p_i(k) E_{\mathbf{x}\sim p_d^i}\Big[log\tilde{p}_D(k|\mathbf{x})\Big]. \tag{6}$$

*The expectation in the last term can be approximated by sampling. Similar to EM mechanism [20] for mixture models, by approximating $-CE_{\mathcal{L}}(\mathbf{p}, \overline{\mathbf{p}})$ with its lower bound, we can perform gradient ascend independently on every sample. Hence, SGD can be applied.*

As shown in (6), in order to facilitate the gradient computation, we substitute cross entropy in (4) by its lower bound and denote this approximate objective function for discriminator by $\widetilde{V}(G, D)$.

## 3.2   The Optimal Discriminator and LLP Classifier

Now, we give the optimal discriminator and the final classifier for LLP based on the analysis of $\widetilde{V}(G, D)$. Firstly, we have the following result of the lower bound in (6).

**Lemma 1.** *The maximization on the lower bound in* (6) *induces an optimal discriminator $D^*$ with a posterior distribution $\tilde{p}_{D^*}(y|\mathbf{x})$, which is consistent with the prior distribution $p_i(y)$ in each bag.*

*Proof.* Taking the aggregation with respect to one bag, for example, the $i^{th}$ bag, we have:

$$E_{\mathbf{x}\sim p_d^i}[log p(\mathbf{x})] = E_{\mathbf{x}\sim p_d^i}log\Big[\frac{p(\mathbf{x}, y)}{\tilde{p}_D(y|\mathbf{x})}\frac{\tilde{p}_D(y|\mathbf{x})}{p(y|\mathbf{x})}\Big] = E_{\mathbf{x}\sim p_d^i}\int p_i(y)log\Big[\frac{p_i(y)p(\mathbf{x}|y)}{\tilde{p}_D(y|\mathbf{x})}\frac{\tilde{p}_D(y|\mathbf{x})}{p(y|\mathbf{x})}\Big]dy$$

$$= E_{\mathbf{x}\sim p_d^i}\int p(y_i)\Big[log\tilde{p}_D(y|\mathbf{x}) + log\frac{p(\mathbf{x}|y)}{p(y|\mathbf{x})}\Big]dy + E_{\mathbf{x}\sim p_d^i}KL(p_i(y)\|\tilde{p}_D(y|\mathbf{x})) \tag{7}$$

$$\geqslant \sum_{k=1}^{K} p_i(k) E_{\mathbf{x}\sim p_d^i}\Big[log\tilde{p}_D(k|\mathbf{x})\Big] + \sum_{k=1}^{K} p_i(k) E_{\mathbf{x}\sim p_d^i}\Big[log\frac{p(\mathbf{x}|k)}{p(k|\mathbf{x})}\Big].$$

Here, because we only consider $\mathbf{x}\sim p_d^i$, $p(\mathbf{x}, y) = p_i(y)p(y|\mathbf{x})$ holds. Note that the last term in (7) is free of the discriminator, and the aggregation can be independently performed within every bag due to the disjoint assumption on bags. Then, maximizing the lower bound in (6) is equivalent to minimizing the expectation of KL-divergence between $p_i(y)$ and $\tilde{p}_D(y|\mathbf{x})$. Because of the infinite capacity assumption on discriminator and the non-negativity of KL-divergence, we have:

$$D^* = \arg\min_D E_{\mathbf{x}\sim p_d^i}KL(p_i(y)\|\tilde{p}_D(y|\mathbf{x})) \Leftrightarrow \tilde{p}_{D^*}(y|\mathbf{x}) \stackrel{a.e.}{=} p_i(y), \mathbf{x}\sim p_d^i(\mathbf{x}). \tag{8}$$

That concludes the proof. □

Lemma 1 tells us that if there is only one bag, then the final classifier $\tilde{p}_{D^*}(y|\mathbf{x}) \stackrel{a.e.}{=} p(y)$. However, there are normally multiple bags in LLP problem, the final classifier will somehow be a trade-off among all the prior proportions $p_i(y), i = 1, 2, \cdots, n$. Next, we will show how the adversarial learning on the discriminator helps to determine the formulation of this trade-off in a weighted aggregation.

**Theorem 1.** *For fixed G, the optimal discriminator $D^*$ for $\widetilde{V}(G, D)$ satisfies:*

$$P_{D^*}(y=k|\mathbf{x}) = \frac{\sum_{i=1}^{n} p_i(k)p_d^i(\mathbf{x})}{\sum_{i=1}^{n} p_d^i(\mathbf{x}) + p_g(\mathbf{x})}, k=1, 2, \cdots, K. \tag{9}$$

*Proof.* According to (4) and (6) and given any generator G, we have:

$$\widetilde{V}(G, D) = \sum_{i=1}^{n} E_{\mathbf{x}\sim p_d^i}\Big[log(1 - P_D(K+1|\mathbf{x}))\Big] + E_{\mathbf{x}\sim p_g}\Big[log P_D(K+1|\mathbf{x})\Big] + \sum_{i=1}^{n}\sum_{k=1}^{K} p_i(k) E_{\mathbf{x}\sim p_d^i}\Big[log\tilde{p}_D(k|\mathbf{x})\Big]$$

$$= \int\Big\{\sum_{i=1}^{n} p_d^i(\mathbf{x})\Big[log\big[\sum_{k=1}^{K} P_D(k|\mathbf{x})\big] + \sum_{k=1}^{K} p_i(k)log\frac{P_D(k|\mathbf{x})}{1 - P_D(K+1|\mathbf{x})}\Big] + p_g(\mathbf{x})log\Big[1 - \sum_{k=1}^{K} P_D(k|\mathbf{x})\Big]\Big\}d\mathbf{x}. \tag{10}$$

By taking the derivative of the integrand, we find the solution in $[0, 1]$ for maximization as (9). □

**Remark 3** (**Beyond the Incontinuity of** $p_g$). *According to [2], the problematic scenario is that the generator is a mapping from a low dimensional space to a high dimensional one. This will result in the density of $p_g(\mathbf{x})$ infeasible. However, based on the definition of $\tilde{p}_D(y|\mathbf{x})$ in (5), we have:*

$$\tilde{p}_{D^*}(y|\mathbf{x}) = \frac{\sum_{i=1}^n p_i(y)p_d^i(\mathbf{x})}{\sum_{i=1}^n p_d^i(\mathbf{x})} = \sum_{i=1}^n w_i(\mathbf{x})p_i(y). \tag{11}$$

*Hence, our final classifier does not depend on $p_g(\mathbf{x})$. Furthermore, (11) explicitly expresses the normalized weights of the aggregation with $w_i(\mathbf{x}) = \frac{p_d^i(\mathbf{x})}{\sum_{i=1}^n p_d^i(\mathbf{x})}$.*

**Remark 4** (**Relationship to One-side Label Smoothing**). *Notice that the optimal discriminator $D^*$ is also related to the one-sided label smoothing mentioned in [28], which is inspirited by [30] and shown to reduce the vulnerability of neural networks to adversarial examples [32].*

*In particular, in our model, we only smooth labels of real data (multi-class) in the discriminator, by setting the targets as the prior proportions $p_i(y)$ in corresponding bags.*

### 3.3 The Objective Function of Generator

Normally, for the generator, we should solve the following optimization problem with respect to $p_g$.

$$\min_G \widetilde{V}(G, D^*) = \min_G E_{\mathbf{x} \sim p_g} log P_{D^*}(K+1|\mathbf{x}). \tag{12}$$

Denoting $C(G) = \max_D \widetilde{V}(G, D) = \widetilde{V}(G, D^*)$, because $\widetilde{V}(G, D)$ is convex in $p_g$ and the supremum of a set of convex functions is still convex, we have the following sufficient and necessary condition of global optimality.

**Theorem 2.** *The global minimum of $C(G)$ is achieved if and only if $p_g = \frac{1}{n}\sum_{i=1}^n p_d^i$.*

*Proof.* Denote $p_d = \sum_{i=1}^n p_d^i$. Hence, according to Theorem 1, we can reformulate $C(G)$ as:

$$C(G) = \sum_{i=1}^n E_{\mathbf{x} \sim p_d^i}\left[log\frac{p_d(\mathbf{x})}{p_d(\mathbf{x})+p_g(\mathbf{x})}\right] + E_{\mathbf{x} \sim p_g}\left[log\frac{p_g(\mathbf{x})}{p_d(\mathbf{x})+p_g(\mathbf{x})}\right] + \sum_{i=1}^n\sum_{k=1}^K p_i(k)E_{\mathbf{x} \sim p_d^i}\left[log\tilde{p}_{D^*}(k|\mathbf{x})\right]$$

$$= 2 \cdot JSD(p_d\|p_g) - 2log(2) - \sum_{i=1}^n E_{\mathbf{x} \sim p_d^i}\left[CE(p_i(y), \tilde{p}_{D^*}(y|\mathbf{x}))\right], \tag{13}$$

where $JSD(\cdot\|\cdot)$ and $CE(\cdot, \cdot)$ are the Jensen-Shannon divergence and cross entropy between two distributions, respectively. However, note that $p_d$ is a summation of $n$ independent distributions, so $\frac{1}{n}p_d$ is a well-defined probabilistic density. Then, we have:

$$C(G^*) = \min_G C(G) = nlog(n) - (n+1)log(n+1) - \sum_{i=1}^n E_{\mathbf{x} \sim p_d^i}\left[CE(p_i(y), \tilde{p}_{D^*}(y|\mathbf{x}))\right] \iff p_{g^*} \stackrel{a.e.}{=} \frac{1}{n}p_d. \tag{14}$$

That concludes the proof. $\qquad\qquad\qquad\qquad\qquad\qquad\qquad\qquad\qquad\qquad\qquad\qquad\qquad\qquad\square$

**Remark 5.** *When there is only one bag, the first two terms in (14) will degenerate as $nlog(n) - (n+1)log(n+1) = -2log2$, which adheres to results in original GANs. On the other hand, the third term manifests the uncertainty on instance label, which is concealed in the form of proportion.*

**Remark 6.** *According to the analysis above, ideally, we can obtain the Nash equilibrium between the discriminator and the generator, i.e., the solution pair $(G^*, D^*)$ satisfies:*

$$\widetilde{V}(G^*, D^*) \geqslant \widetilde{V}(G^*, D), \forall D; \widetilde{V}(G^*, D^*) \leqslant \widetilde{V}(G, D^*), \forall G. \tag{15}$$

However, as shown in [8], a well-trained generator would lead to the inefficiency of supervised information. In other words, the discriminator would possess the same generalization ability as merely training it on $L_{prop}$. Hence, we apply feature matching (FM) to the generator and obtain its alternative objective by matching the expected value of the features (statistics) on an intermediate layer of the discriminator [28]: $L(G) = \|E_{\mathbf{x} \sim \frac{1}{n}p_d}f(\mathbf{x}) - E_{\mathbf{x} \sim p_g}f(\mathbf{x})\|_2^2$. In fact, FM is similar to the perceptual loss for style transfer in a concurrent work [15], and the goal of this improvement is to impede the "perfect" generator resulting in unstable training and discriminator with low generalization.

### 3.4 LLP-GAN Algorithm

So far, we have clarified the objective functions of both discriminator and generator in LLP-GAN. When accomplishing the training stage, the discriminator can be put into effect as the final classifier.

The strict proof for algorithm convergence is similar to that in [11]. Because $\max_D \widetilde{V}(G, D)$ is convex in $G$, and the subdifferential of $\max_D \widetilde{V}(G, D)$ contains that of $\widetilde{V}(G, D^*)$ in every step, the line search method (stochastic) gradient descent converges [7].

We present the LLP-GAN algorithm, which coincides with the algorithm of the original GAN [11].

---

**Algorithm 1:** LLP-GAN Training Algorithm

---

**Input**: The training set $\mathcal{L} = \{(\mathcal{B}_i, \mathbf{p}_i)\}_{i=1}^n$; $L$: number of total iterations; $\lambda$: weight parameter.
**Output**: The parameters of the final discriminator $D$.
Set $m$ to the total number of training data points.
**for** $i=1:L$ **do**

    Draw $m$ samples $\{\mathbf{z}^{(1)}, \mathbf{z}^{(2)}, \cdots, \mathbf{z}^{(m)}\}$ from a simple-to-sample noise prior $p(\mathbf{z})$ (e.g., $N(0, I)$).
    Compute $\{G(\mathbf{z}^{(1)}), G(\mathbf{z}^{(2)}), \cdots, G(\mathbf{z}^{(m)})\}$ as sampling from $p_g(\mathbf{x})$.

    Fix the generator $G$ and perform gradient ascent on parameters of $D$ in $\widetilde{V}(G, D)$ for one step.
    Fix the discriminator $D$ and perform gradient descent on parameters of $G$ in $L(G)$ for one step.
**end**
Return parameters of the discriminator $D$ in the last step.

---

## 4 Experiments

Four benchmark datasets, MNIST, SVHN, CIFAR-10, and CIFAR-100 are investigated in our experiments[1]. In addition to test error comparison, three issues are discussed: the generated samples, the performance under different selections of hyperparameter $\lambda$, and the algorithm scalability.

### 4.1 Experimental Setup

To keep up the same settings in previous work, bag size is fixed as 16, 32, 64, and 128. We divide training data into bags. MNIST data can be found in the code in the Supplementary Material. We conceal the accessible instance-level labels by replacing them with bag-level label proportions. Note that we still need the instance-level labels in test data to justify the effectiveness of the obtained classifier.

### 4.2 Results on CIFAR-10

Firstly, we perform both DLLP and LLP-GAN on CIFAR-10, which is a computer-vision dataset used for object recognition with 60,000 color images belonging to 10 categories, respectively. In the experimental setting, the training data is equally divided into five minibatches, with 10,000 images in each one, and the test data with exactly 1,000 images in every category.

#### 4.2.1 Convergence Analysis

We report the convergence curves of test error (y-axis) with respect to the epoch (x-axis) under different bag sizes in Figure 3. As shown, our results are highly superior to DLLP in most of the epochs, with significant convergence in test error. In contrast, DLLP fails to converge under relatively large bag sizes (i.e., 64 and 128). Also, our method achieves a better performance in accuracy.

#### 4.2.2 Generated Samples

The original GAN suffers from inefficient training on the generator [2]. It suggests that the discriminator and generator cannot simultaneously perform well [8]. In LLP-GAN, although it is the discriminator that we are interested in, we still expect a competent generator to construct efficient adversarial learning paradigm. As a result, we look at the generated samples of original GANs with

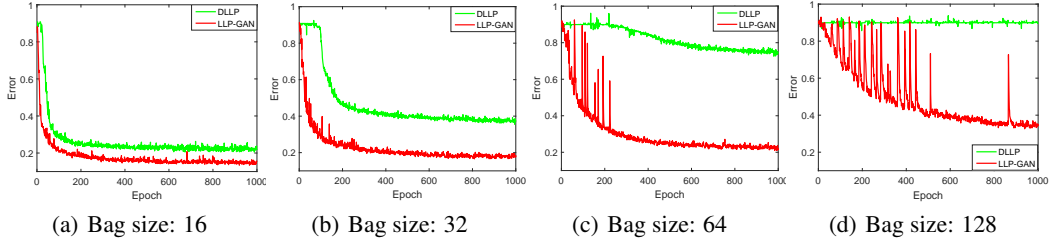

| (a) Bag size: 16 | (b) Bag size: 32 | (c) Bag size: 64 | (d) Bag size: 128 |

Figure 3: The convergence curves on CIFAR-10 w/ different bag sizes.

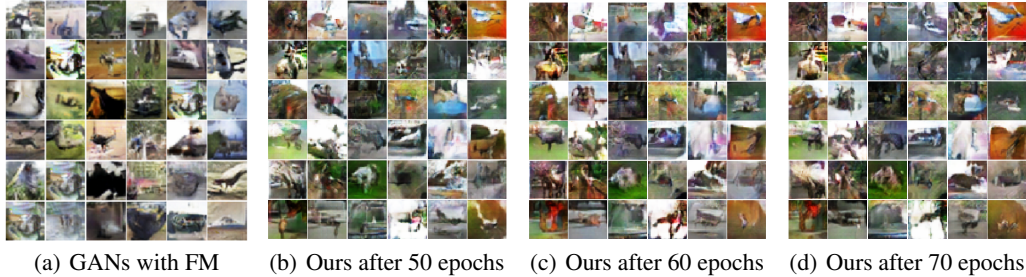

| (a) GANs with FM | (b) Ours after 50 epochs | (c) Ours after 60 epochs | (d) Ours after 70 epochs |

Figure 4: Generated samples on CIFAR-10.

FM in Figure 4(a) and our method in Figure 4(b), 4(c) and 4(d). It demonstrates that our approach can stably learn a comparable generator to produce similar samples to that of GANs.

## 4.3 The Results of Error Rate

Secondly, DLLP and LLP-GAN are carried out on four benchmark datasets with different bag sizes in Table 1. We also give the fully supervised learning results as the baselines. In detail, baseline for MNIST and CIFAR-10 is offered by [25]. We describe its architecture in the Supplementary Material. Network in [18] is used as the baseline for SVHN and CIFAR-100.

In terms of test error, our method reaches a relatively better result, except for the simplest task MNIST, where both algorithms can attain satisfying results. However, DLLP becomes unacceptable when the bag size increases, while our method can properly tackle relatively large bag size. Besides, for each

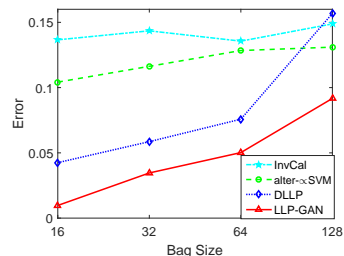

Figure 5: The average error rates w/ different bag sizes.

dataset, LLP becomes extremely difficult as bag size soaring, which is consistent with our intuition. Again, the architectures of our network are given in the Supplementary Material.

Table 1: Test error rates (%) on benchmark datasets w/ different bag sizes.

| Dataset | Algorithm | Bag Size | | | | Baseline |
| | | 16 | 32 | 64 | 128 | CNNs |
| --- | --- | --- | --- | --- | --- | --- |
| MNIST | DLLP | 1.23 (0.100) | 1.33 (0.094) | 1.57 (0.088) | 3.55 (0.27) | 0.36 |
| | LLP-GAN | **1.10** (0.026) | **1.23** (0.088) | **1.40** (0.089) | **3.49** (0.27) | |
| SVHN | DLLP | 4.45 (0.069) | 5.29 (0.54) | 5.80 (0.91) | 39.73 (1.60) | 2.35 |
| | LLP-GAN | **4.03** (0.021) | **4.83** (0.51) | **5.42** (0.59) | **11.17** (1.12) | |
| CIFAR-10 | DLLP | 19.70 (0.77) | 34.39 (0.82) | 68.32 (1.34) | 82.89 (2.66) | 9.27 |
| | LLP-GAN | **13.68** (0.35) | **16.23** (0.43) | **21.03** (1.82) | **27.39** (4.31) | |
| CIFAR-100 | DLLP | 53.24 (0.77) | 98.38 (0.11) | 98.65 (0.09) | 98.98 (0.08) | 35.68 |
| | LLP-GAN | **50.95** (0.67) | **56.44** (0.78) | **64.37** (1.52) | **85.01** (1.81) | |

Because InvCal and alter-$\propto$SVM are originally designed for binary problem, we randomly select two classes and merely conduct binary classification on all datasets. The detailed results are provided in the Supplementary Material. The average error rates with different bag sizes are displayed in Figure 5. From the results, we can confidently tell the advantage of our algorithm in performance,

especially when the bag size is relatively large. Indeed, at this moment, we cannot clarify to what extent this advantage attributes to the deep learning model. However, in spite of both using deep learning models, our method constantly performs better than DLLP.

## 4.4   Hyperparameter Analysis and Complexity with Sample Size

Thirdly, we illustrate the convergence curves of MNIST, SVHN, and CIFAR-10 under different $\lambda$s in Figure 6(a), 6(b) and 6(c). For simpler task (MNIST), the performance is not sensitive to $\lambda$. However, for harder task (CIFAR-10), the performance becomes sensitive to $\lambda$. On the other hand, smaller $\lambda$ demonstrates more fluctuations, which is much severer in simpler tasks (MNIST and SVHN). Besides, Figure 6(b) indicates that the convergence speed may be sensitive to the choice of $\lambda$. In most of the cases, $\lambda \geqslant 1$ is a good choice, leading to a comparable performance within limited training time.

In addition, fixing the bag size, we provide the relative training time (training time per bag) to the relative sample size in Figure 6(d). We take logarithmic operation on sample size (x-axis). It demonstrates that the relative training time is asymptotically linear to the logarithmic sample size $m$. Denote the total training time as $t$, then $t \approx O(m\mathbf{ln}m) < O(m^2)$. Here, we assume that sample size and # of bags are with same magnitude, due to the relative small bag sizes involved in our study.

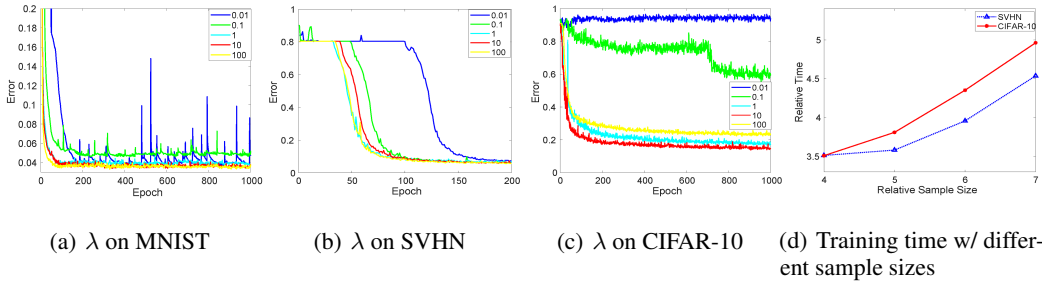

(a) $\lambda$ on MNIST        (b) $\lambda$ on SVHN        (c) $\lambda$ on CIFAR-10        (d) Training time w/ different sample sizes

Figure 6: Analysis on hyperparameter and complexity.

## 4.5   Discussion on Experimental Results

Two issues should be clarified for experiments. Firstly, as shown in Figure 3, the results demonstrate oscillation as bag size soaring. This phenomenon indicates a common drawback of deep models: For more complex objective surfaces (more possible label candidates), normally the convergence will be dramatically getting worse, due to more chances to attain local minima or saddle points of the objective. Secondly, because our results are based on original datasets without data augmentation, the reported DLLP performance is worse than that in the concurrent [10].

## 5   Conclusion

This paper proposed a new algorithm LLP-GAN for LLP problem in virtue of the adversarial learning based on GANs. Consequently, our method is superior to existing methods in the following three aspects. Firstly, it demonstrates nice theoretical properties that are innately in accordance with GANs. Secondly, LLP-GAN can produce a probabilistic classifier, which benefits from the generative model and meets the proportion consistency naturally. Thirdly, on account of equipping CNNs, our algorithm is suitable for the large-scale problem, especially for image datasets. Additionally, the experiments on four benchmark datasets have verified all these advantages of our approach.

Nevertheless, limitations in our method can be summarized in four aspects. Firstly, learning complexity in the sense of PAC has not been involved in this study. That is to say, we cannot evaluate the performance under limited data. Secondly, there is no guarantee on algorithm robustness to data perturbations, notably when the proportions are imprecisely provided. Thirdly, varying GAN models (such as WGAN [3]) are not fully considered, and their performance is still unknown. In addition, in many real-world applications, the bags are built based on certain features, such as the education levels and job titles, rather than randomly established. Hence, a practical issue will be to ensure good performance under these non-random bag assignments. To overcome these drawbacks will shed light on the promising improvement of our current work.

## Acknowledgements

This work is supported by grants from: National Natural Science Foundation of China (No.61702099, 71731009, 61472390, 71932008, 91546201, and 71331005), Science and Technology Service Network Program of Chinese Academy of Sciences (STS Program, No.KFJ-STS-ZDTP-060), and the Fundamental Research Funds for the Central Universities in UIBE (No.CXTD10-05). Bo Wang would like to acknowledge that this research was conducted during his visit at Texas A&M University and thank Dr. Xia Hu for his hosting and insightful discussions.

## Footnotes

[1]Code is available at https://github.com/liujiabin008/LLP-GAN.

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
