[Supplementary Material 1]

# Learning from Label Proportions with Generative Adversarial Network

Jiabin Liu[1], Bo Wang[2], Zhiquan Qi[1], Yingjie Tian[1], and Yong Shi[1]

[1]University of Chinese Academy of Sciences

[2]University of International Business and Economics

December 5, 2019

# Outline

# Outline

# Problem Description (1)

From supervised learning to weakly supervised learning:

- Combating over-fitting issue: e.g., big data

- The lack of fully supervised data: infeasible or labor-intensive[1]

- Certain constraints: e.g., privacy[2]

- The ubiquity of weakly labeled learning (WeLL): Semi-supervised learning (SSL) and Multi-instance Learning (MIL)

- Learning with bags: MIL and learning from label proportions (LLP)

Figure: An illustration of multi-class LLP.

- The data belongs to three categories and is partitioned into four non-overlapping groups.
- The sizes of green, blue, and orange rectangles respectively denote available label proportions in different categories.
- We only know feature information and class proportions.

# Challenges

- The uncertainty in label inference (proportional information in bags)

- Strict assumption on data distribution (statistical approaches, e.g., MeanMap[3] and Laplacian MeanMap[4])

- NP-hard combinatorial optimization issue (SVM-based methods, e.g., InvCal[5] and alter-$\propto$SVM[6])

- The lack of scalability (shallow models)

[3] N. Quadrianto, A. J. Smola, T. S. Caetano, et al. "Estimating labels from label proportions". In: *Journal of Machine Learning Research* 10.Oct (2009), pp. 2349–2374.

[4] G. Patrini et al. "(Almost) no label no cry". In: *Advances in Neural Information Processing Systems*. 2014, pp. 190–198.

[5] S. Rueping. "SVM classifier estimation from group probabilities". In: *International Conference on Machine Learning*. pp. 911–918.

[6] F. X. Yu, D. Liu, S. Kumar, et al. "$\propto$-SVM for learning with label proportions". In: *International Conference on Machine Learning*. 2013, pp. 504–512.

In this paper, we apply GANs to LLP in large scale scenarios.

- GAN is an elegant recipe for solving WeLL problem[7].

- Generative models offer explicit or implicit representations for WeLL[8].

- LLP-GAN is free of strict assumptions through the adversarial scheme.

# Contributions

- We propose a simple improvement based on entropy regularization for the existing deep LLP solver.
- We reveal relationship between prior class proportions and posterior class likelihoods.
- We offer a decomposition representation of the class likelihood with respect to the prior class proportions, which verifies the existence of the final classifier.
- We empirically show that our method can achieve SOTA performance on large-scale LLP problems with a low computational complexity.

# Outline

# Problem Settings

- All the bags are disjoint and let $\mathcal{B}_i = \{\mathbf{x}_i^1, \mathbf{x}_i^2, \cdots, \mathbf{x}_i^{N_i}\}, i = 1, 2, \cdots, n$ be the bags in training set.

- Training data is $\mathcal{D} = \mathcal{B}_1 \cup \mathcal{B}_2 \cup \cdots \cup \mathcal{B}_n, \mathcal{B}_i \cap \mathcal{B}_j = \emptyset, \forall i \neq j$, where the total number of bags is $n$.

- Assuming we have $K$ classes, for $\mathcal{B}_i$, let $\mathbf{p}_i$ be a $K$-element vector, where the $k^{th}$ element $p_i^k$ is the proportion of instances belonging to the class $k$, with the constraint $\sum_{k=1}^K p_i^k = 1$, i.e.,

$$p_i^k := \frac{|\{j \in [1 : N_i] | \mathbf{x}_i^j \in \mathcal{B}_i, y_i^{j*} = k\}|}{|\mathcal{B}_i|}. \tag{1}$$

Here, $[1 : N_i] = \{1, 2, \cdots, N_i\}$ and $y_i^{j*}$ is the unaccessible ground-truth instance-level label of $\mathbf{x}_i^j$.

# Deep LLP Appoach

- Suppose that $\tilde{\mathbf{p}}_i^j = p_\theta(\mathbf{y}|\mathbf{x}_i^j)$ is the vector-valued DNNs output for $\mathbf{x}_i^j$, where $\theta$ is the network parameter.

- The bag-level label proportion in the $i^{th}$ bag is to incorporate the element-wise posterior probability:

$$\bar{\mathbf{p}}_i = \frac{1}{N_i} \bigoplus_{j=1}^{N_i} \tilde{\mathbf{p}}_i^j = \frac{1}{N_i} \bigoplus_{j=1}^{N_i} p_\theta(\mathbf{y}|\mathbf{x}_i^j). \tag{2}$$

- Entropy Regularization for DLLP[9]:

$$L = L_{prop} + \lambda E_{in} = -\sum_{i=1}^{n} \mathbf{p}_i^\mathsf{T} log(\bar{\mathbf{p}}_i) - \lambda \sum_{i=1}^{n} \sum_{j=1}^{N_i} (\tilde{\mathbf{p}}_i^j)^\mathsf{T} log(\tilde{\mathbf{p}}_i^j). \tag{3}$$

# Outline

We illustrate the LLP-GAN framework as follows.

Figure: An illustration of our LLP-GAN framework.

# The Objective of Discriminator (1)

- We normalize the first $K$ classes in $P_D(\cdot|\mathbf{x})$ as instance-level posterior probability $\tilde{p}_D(\cdot|\mathbf{x})$ and compute $\overline{\mathbf{p}}$ based on (2).
- The *ideal* optimization problem for the discriminator of LLP-GAN is:

$$\max_D \quad V(G,D) = L_{unsup} + L_{sup} = L_{real} + L_{fake} - \lambda CE_{\mathcal{L}}(\mathbf{p}, \overline{\mathbf{p}})$$

$$= \sum_{i=1}^{n} E_{\mathbf{x} \sim p_d^i}\Big[\log P_D(y \leq K|\mathbf{x})\Big] + E_{\mathbf{x} \sim p_g}\Big[\log P_D(K+1|\mathbf{x})\Big] + \lambda \sum_{i=1}^{n} \mathbf{p}_i^{\mathsf{T}} \log(\overline{\mathbf{p}}_i).$$

$$(4)$$

Here, $p_g(\mathbf{x})$ is the distribution of the synthesized data.

- The Lower Bound Approximation:

$$- CE_{\mathcal{L}}(\mathbf{p}, \overline{\mathbf{p}}) = \sum_{i=1}^{n}\sum_{k=1}^{K} p_i(k) log\Big[\frac{1}{N_i}\sum_{j=1}^{N_i}\tilde{p}_D(k|\mathbf{x}_i^j)\Big]$$

$$\simeq \sum_{i=1}^{n}\sum_{k=1}^{K} p_i(k) log\Big[\int p_d^i(\mathbf{x})\tilde{p}_D(k|\mathbf{x})d\mathbf{x}\Big] \geqslant \sum_{i=1}^{n}\sum_{k=1}^{K} p_i(k) E_{\mathbf{x}\sim p_d^i}\Big[log\,\tilde{p}_D(k|\mathbf{x})\Big].$$

$$(5)$$

- The expectation in the last term can be approximated by sampling. Similar to EM mechanism[10] for mixture models, by approximating $-CE_{\mathcal{L}}(\mathbf{p}, \overline{\mathbf{p}})$ with its lower bound, we can perform gradient ascend independently on every sample, e.g., SGD.

# The Optimity

## Lemma

*The maximization on lower bound in* (5) *induces optimal discriminator $D^*$ with a posterior distribution $\tilde{p}_{D^*}(y|\mathbf{x})$, which is consistent with the prior distribution $p_i(y)$ in each bag.*

## Theorem

*For fixed G, the optimal discriminator $D^*$ for $\widetilde{V}(G, D)$ satisfies:*

$$P_{D^*}(y = k|\mathbf{x}) = \frac{\sum_{i=1}^{n} p_i(k) p_d^i(\mathbf{x})}{\sum_{i=1}^{n} p_d^i(\mathbf{x}) + p_g(\mathbf{x})}, \, k = 1, 2, \cdots, K. \quad (6)$$

- The generator is a mapping from a low dimensional space to a high dimensional one.
- The density of $p_g(\mathbf{x})$ is infeasible[11].
- Based on the definition of $\tilde{p}_D(y|\mathbf{x})$, we have:

$$\tilde{p}_{D^*}(y|\mathbf{x}) = \frac{\sum_{i=1}^{n} p_i(y) p_d^i(\mathbf{x})}{\sum_{i=1}^{n} p_d^i(\mathbf{x})} = \sum_{i=1}^{n} w_i(\mathbf{x}) p_i(y). \tag{7}$$

- Our final classifier does not depend on $p_g(\mathbf{x})$.
- (7) explicitly expresses the normalized weights of the aggregation with $w_i(\mathbf{x}) = \frac{p_d^i(\mathbf{x})}{\sum_{i=1}^{n} p_d^i(\mathbf{x})}$.

- Normally, we should solve the following optimization problem with respect to $p_g$ for the generator.

$$\min_G \widetilde{V}(G, D^*) = \min_G E_{\mathbf{x} \sim p_g} log P_{D^*}(K+1|\mathbf{x}). \qquad (8)$$

- However, a well-trained generator would lead to the inefficiency of supervised information.

- Hence, we apply feature matching (FM) to the generator and obtain its alternative objective by matching the expected value of features (statistics) on an intermediate layer of the discriminator:

$$L(G) = \|E_{\mathbf{x} \sim \frac{1}{n} p_d} f(\mathbf{x}) - E_{\mathbf{x} \sim p_g} f(\mathbf{x})\|_2^2 \qquad (9)$$

**Algorithm 1:** LLP-GAN Training Algorithm

**Input:** The training set $\mathcal{L} = \{(\mathcal{B}_i, \mathbf{p}_i)\}_{i=1}^n$; $L$: number of total iterations; $\lambda$: weight parameter.

**Output:** The parameters of the final discriminator $D$.

Set $m$ to the total number of training data points.

**for** $i=1:L$ **do**

    Draw $m$ samples $\{\mathbf{z}^{(1)}, \mathbf{z}^{(2)}, \cdots, \mathbf{z}^{(m)}\}$ from a simple-to-sample noise prior $p(\mathbf{z})$ (e.g., $N(0, I)$).

    Compute $\{G(\mathbf{z}^{(1)}), G(\mathbf{z}^{(2)}), \cdots, G(\mathbf{z}^{(m)})\}$ as sampling from $p_g(\mathbf{x})$.

    Fix the generator $G$ and perform gradient ascent on parameters of $D$ in $\widetilde{V}(G, D)$ for one step.

    Fix the discriminator $D$ and perform gradient descent on parameters of $G$ in $L(G)$ for one step.

**end**

Return parameters of the discriminator $D$ in the last step.

# Outline

DLLP v.s. LLP-GAN (proposed)

(a) Bag size: 16      (b) Bag size: 32      (c) Bag size: 64      (d) Bag size: 128

Figure: The convergence curves on CIFAR-10 w/ different bag sizes.

# Generated Samples

To validate the effectiveness of generator in LLP-GAN, we compare the generated samples of our model with that of the standard GAN with feature matching.

(a) GANs with FM    (b) 50 epochs (ours)    (c) 60 epochs (ours)    (d) 70 epochs (ours)

Figure: Generated samples on CIFAR-10.

(a) $\lambda$ on MNIST    (b) $\lambda$ on SVHN    (c) $\lambda$ on CIFAR-10

Figure: Analysis on hyperparameter.

# Error Rates Comparison(1)

The results are the average performances of four datasets: MNIST, SVHN, CIFAR-10, and CIFAR-100.

Figure: The average error rates w/ different bag sizes.

Table: Test error rates (%) on benchmark datasets w/ different bag sizes.

| Dataset | Algorithm | Bag Size | | | | Baseline CNNs |
|---|---|---|---|---|---|---|
| | | 16 | 32 | 64 | 128 | |
| MNIST | DLLP | 1.23 (0.100) | 1.33 (0.094) | 1.57 (0.088) | 3.55 (0.27) | 0.36 |
| | LLP-GAN | **1.10** (0.026) | **1.23** (0.088) | **1.40** (0.089) | **3.49** (0.27) | |
| SVHN | DLLP | 4.45 (0.069) | 5.29 (0.54) | 5.80 (0.91) | 39.73 (1.60) | 2.35 |
| | LLP-GAN | **4.03**(0.021) | **4.83**(0.51) | **5.42**(0.59) | **11.17**(1.12) | |
| CIFAR-10 | DLLP | 19.70 (0.77) | 34.39 (0.82) | 68.32 (1.34) | 82.89 (2.66) | 9.27 |
| | LLP-GAN | **13.68** (0.35) | **16.23** (0.43) | **21.03** (1.82) | **27.39** (4.31) | |
| CIFAR-100 | DLLP | 53.24(0.77) | 98.38(0.11) | 98.65(0.09) | 98.98(0.08) | 35.68 |
| | LLP-GAN | **50.95**(0.67) | **56.44**(0.78) | **64.37**(1.52) | **85.01**(1.81) | |

Table: Binary test error rates (%) on benchmark datasets w/ different bag sizes.

| Dataset | Algorithm | Bag Size | | | |
|---------|-----------|------|------|------|------|
| | | 16 | 32 | 64 | 128 |
| MNIST | InvCal | 0.50 | 0.55 | 1.25 | 0.1 |
| | alter-pSVM | 0.20 | 0.20 | 0.25 | 0.2 |
| | DLLP | 0.049 | 0.049 | 0.049 | 0.049 |
| | LLP-GAN | **0.047** | **0.047** | **0.047** | **0.047** |
| CIFAR-10 | InvCal | 28.95 | 29.16 | 26.47 | 31.84 |
| | alter-pSVM | 24 | 26.74 | 30.32 | 27.95 |
| | DLLP | 11.31 | 15.83 | 18.96 | 22.59 |
| | LLP-GAN | **1.39** | **1.61** | **11.59** | **18.29** |
| SVHN | InvCal | 11.55 | 13.35 | 12.95 | 12.70 |
| | alter-pSVM | 7.05 | 7.95 | 7.95 | 11.15 |
| | DLLP | **1.38** | **1.7** | 3.77 | 24.45 |
| | LLP-GAN | 1.49 | 1.8 | **3.46** | **9.23** |

# Error Rates Comparison (4)

(a) MINST

(b) SVHN

(c) CIFAR-10

(d) CIFAR-100

Figure: Multi-class test error rates (%) on benchmark w/ different bag sizes.

# Outline

# Conclusion

- This paper proposed a new algorithm LLP-GAN for LLP problem in virtue of the adversarial learning based on GANs.

- Our method is superior to existing methods in three aspects.

    - Nice theoretical properties

    - A probabilistic classifier

    - Scalability: e.g., image data applications

# Future Work

- Learning complexity in the sense of PAC is not involved in this study.

- There is no guarantee on algorithm robustness to data perturbations: e.g., imprecise proportions.

- Varying GANs are not considered in our current model and their performance is unknown: e.g. WGAN[12].

- The performance of LLP-GAN on tabular data and structured (non-random) data[13] is not included.

# Acknowledgement

- This work is supported by grants from: National Natural Science Foundation of China (No.61702099,71731009, and 61472390), Science and Technology Service Network Program of Chinese Academy of Sciences (STS Program, No.KFJ-STS-ZDTP-060), and the Fundamental Research Funds for the Central Universities in UIBE (No.CXTD10-05).

- Bo Wang would like to acknowledge that this research was conducted during his visit at Texas A&M University and thank Dr. Xia Hu for his hosting and insightful discussions.

- We appreciate the dedicated reviewers and area chair for their valuable insights in the reviews and meta review to help improve this paper.

## Footnotes

[1] Z. Wang and J. Feng. "Multi-class learning from class proportions". In: *Neurocomputing* 119.16 (2013),

[2] Z. Qi, B. Wang, F. Meng, et al. "Learning with label proportions via NPSVM". In: *IEEE Transactions on Cybernetics* 47.10 (2017), pp. 3293–3305.

[7] T. Salimans, I. Goodfellow, W. Zaremba, et al. "Improved techniques for training GANs". In: *Advances in Information Processing Systems*. 2016, pp. 2234–2242.

[8] D. P. Kingma and M. Welling. "Auto-encoding variational bayes". In: *arXiv preprint arXiv:1312.6114* (2013).

[9] E. M. Ardehaly and A. Culotta. "Co-training for demographic classification using deep learning from label proportions". *International Conference on Data Mining Workshops.* IEEE. 2017, pp. 1017–1024.

[10] T. K. Moon. "The expectation-maximization algorithm". In: *IEEE Signal processing magazine* 13.6 (1996), pp. 47–60.

[11]M. Arjovsky and L. Bottou. "Towards principled methods for training generative adversarial networks". In: *International Conference on Learning Representations*. 2016.

[12] M. Arjovsky, S. Chintala, and L. Bottou. "Wasserstein generative adversarial networks". In: *ICML*. 2017, pp. 214–223.

[13] G. Patrini et al. "(Almost) no label no cry". In: *Advances in Neural Information Processing Systems*. 2014, pp. 190–198.


[Supplementary Material 2]

# 1 The Architectures of Networks

In Table 1, we deliver the CNNs architecture used in our experiments. We describe the operation in the format of "filter size / type / # of output channel". Note that the network architecture of LLP-GAN for SVHN and CIFAR-100 is the same as that for CIFAR-10. In particular, *Dense* means fully connected layer. *Transpose_conv* is the deconvolution layer. In our model, we choose ReLU as the activation function.

Following a standard setting in the previous work [1, 4], we perform 11-way softmax on the 10-dimensional output in the last fully connected layer. In detail, we add an extra dimension to the output and fix its value as zero, which is a form of over-parameterization. Then, we apply 11-way softmax to the 11-dimensional vector.

Table 1: Network architectures in LLP-GAN.

| Generator | | Discriminator | |
|---|---|---|---|
| MNIST | CIFAR-10 | MNIST | CIFAR-10 |
| Input 28×28 or 32×32 monochrome or RGB image | | | |
| | | | dropout 0.2 |
| Dense-BN 500 | Dense-BN 4*4*512 | 5×5 conv. 32 | 3×3 conv. 64 |
| | | 3×3 conv. 64 | 3×3 conv. 64 |
| | | | 3×3 conv. 64 |
| | | | dropout 0.5 |
| Dense-BN 500 | 5×5 Transpose_conv-BN 256 | 1×1 conv. 32 | 3×3 conv. 128 |
| | | | 3×3 conv. 128 |
| | | | 3×3 conv. 128 |
| | | | dropout 0.5 |
| Dense-BN 784 | 5×5 Transpose_conv-BN 128 | Dense 1024 | 3×3 conv. 256 |
| | | | 1×1 conv. 128 |
| | | | 1×1 conv. 64 |
| | | | global meanpooling 8 |
| | 5×5 Transpose_conv 3 | Dense 10 | Dense 10 |
| | | 11-way softmax (over-parameterization) | |

The CNNs architectures used in the baselines for MNIST and CIFAR-10 are given in Table 2, which are the same as that in [3]. Besides, the CNNs used in the baselines for SVHN and CIFAR-100 are the same as that in [2].

Table 2: The baseline's architectures.

| MNIST | CIFAR-10 |
|---|---|
| Input 28×28 or 32×32 monochrome or RGB image | |
| 5×5 conv. ReLU 32 | 3×3 conv. BN LeakyReLU 96 |
| | 3×3 conv. BN LeakyReLU 96 |
| | 3×3 conv. BN LeakyReLU 96 |
| 2×2 max-pooling stride 2 BN | 2×2 max-pooling stride 2 BN |
| 3×3 conv. BN ReLU 64 | 3×3 conv. BN LeakyReLU 192 |
| 3×3 conv. BN ReLU 64 | 3×3 conv. BN LeakyReLU 192 |
| | 3×3 conv. BN LeakyReLU 192 |
| 2×2 max-pooling stride 2 BN | 2×2 max-pooling stride 2 BN |
| 3×3 conv. 128 BN ReLU | 3×3 conv. BN LeakyReLU 192 |
| | 1×1 conv. BN LeakyReLU 192 |
| 1×1 conv. BN ReLU 10 | 1×1 conv. BN LeakyReLU 10 |
| global meanpool BN | global meanpool BN |
| Dense-BN 10 | |
| 10-way softmax | |

# 2 More results on Performance

## 2.1 Binary Case

In addition to the comparison between DLLP and LLP-GAN, we investigate results of other two representative LLP solvers: InvCal and alter-$\propto$SVM. Because they are originally designed for binary problem, we randomly select two classes and merely conduct binary classification on all datasets with four algorithms. The comparison on test error rates is displayed in Table 3.

Table 3: Binary test error rates (%) on benchmark datasets with different bag sizes.

| Dataset | Algorithm | Bag Size | | | |
|---|---|---|---|---|---|
| | | 16 | 32 | 64 | 128 |
| MNIST | InvCal | 0.50 | 0.55 | 1.25 | 0.1 |
| | alter-pSVM | 0.20 | 0.20 | 0.25 | 0.2 |
| | DLLP | 0.049 | 0.049 | 0.049 | 0.049 |
| | LLP-GAN | **0.047** | **0.047** | **0.047** | **0.047** |
| CIFAR-10 | InvCal | 28.95 | 29.16 | 26.47 | 31.84 |
| | alter-pSVM | 24 | 26.74 | 30.32 | 27.95 |
| | DLLP | 11.31 | 15.83 | 18.96 | 22.59 |
| | LLP-GAN | **1.39** | **1.61** | **11.59** | **18.29** |
| SVHN | InvCal | 11.55 | 13.35 | 12.95 | 12.70 |
| | alter-pSVM | 7.05 | 7.95 | 7.95 | 11.15 |
| | DLLP | **1.38** | **1.7** | 3.77 | 24.45 |
| | LLP-GAN | 1.49 | 1.8 | **3.46** | **9.23** |

## 2.2 Multi-class Case

We report multi-calss error rates with the standard deviations of DLLP and LLP-GAN on benchmark datasets in Figure 1. It is based on the results in Table 1 of our paper.

(a) MNIST

(b) SVHN

(c) CIFAR-10

(d) CIFAR-100

Figure 1: Multi-class test error rates (%) on benchmark datasets with different bag sizes.

## 2.3 DLLP with Entropy Regularization

Although DLLP with Entropy Regularization is a side contribution of our work, as claimed in the paper, we consider not to include it as a baseline. The reason is the experimental results suggest that the original DLLP has already converged to the solution with fairly low instance-level entropy, which makes the regularization term redundant. We demonstrate this statement in Figure 2.

Figure 2: Sum of instance-level entropy on MNIST.

## 2.4 The Randomness of Bag Assignment

The distribution of proportions has an huge impact on LLP algorithm performance. Hence, fixing bag size, we randomly construct bags for multiple times and present the accuracy performance in Table 2.4. The result shows the stability of our method. Currently, we can only artificially build LLP datasets from supervised ones. However, the gap between the importance of LLP in real-life and lack of specific LLP datasets exactly suggests the meaning of our work: It is worthy of devoting efforts to further study in order to draw more attention from the community.

Table 4: The performance on accuracy with deviation under multiple random bag generations on MNIST. (Due to the time limitation, # of random are differently chosen.)

| Bag Size (# of Random) | # of Errors | Accuracy (%) (Deviation) | Baseline (CNN) |
|---|---|---|---|
| 16 (7) | 106 | 98.94 (0.0285) | |
| 32 (22) | 124 | 98.76 (0.0542) | |
| 64 (45) | 147 | 98.53 (0.11) | 99.64 |
| 128 (85) | 335 | 96.65 (0.4) | |