[Reviews · NeurIPS 2019]

Reviewer 1



1. motivation of the method. I do not quite understand the motivation of the work. Although the method gives better empirical performance, why is the GAN objective necessary? This is not justified either intuitively or theoretically. 2. a missing baseline In Section 2.3, the "entropy regularization for LLP" appears to be a plausible idea. It adds a low entropy term to encourage high labeling confidence. The authors presented this section but "will not look at the performacne of this extenion in this paper". This is strange, and I think this should be an obvious baseline in the experiment. 3. details of the optimization After Remark I: "it is easier to perform the gradient method on the lower bound, because it swaps the order of log and the summation operation." I cannot agree with the argument here. All popular deep learning packages are based on automatic differentiation. Implementing one is not more difficulty than the other. -----after rebuttal and discussion --- Most of my concerns have been addressed in the rebuttal. I suggest adding the new results in the final paper.

Reviewer 2



Summary: The paper proposes to use GANs in the LLP setting, where only the proportions are known per bag of covariates (say images), the goal is to create a classifier on the covariate level. Similar to the previous work in semi-supervised learning with GANs, the discriminator in this case classifies between the current classes and a new class for fake samples. Here, the loss function like the normal GANs uses two term, one for the supervision for the matching of the label proportion, the other for the adversarial training. The authors then proceed to analyse this loss function. Originality: I have not previously seen the use of GANs in the LLP setting or the use of the lower bound approximation that allows for SGD on individuals. Also, I think the work done on analysis of the loss function is also new and interesting, however I am not an expert on GANs. Clarity: I think the paper is mostly well written, however I have some comments and questions below. Experiments: The authors compares their methodology to the current literature baseline that does not employ GANs, and also other baselines that are available for binary classification. In particular, they verify their algorithm on MNIST, SVNH, CIFAR10 and CIFAR100 for different bag size beats the current baselines. I have some question on experiments regarding the CIFAR100 experiment below. Comments/Questions 1. Can authors give intuition on why the use of GANs would be beneficial in the setting of the LLP, the motivation was not very clear in the paper. 2. Since the discriminator constructed at the end is a K+1 class (including the fake data class) classifier, what do authors do for true K class when you use it? 3. How would the performance change if you were to use SGD on bags instead of individuals for LLP GAN? It might be an interesting experiment to run. 4. Is the current DLLP baseline implemented with SGD on bags or uses also the lower bound approximation? 5. Figure 3 seems to have unstable behavior during training, is this expected behavior? 6. In appendix Figure 1 for the multi-class case, the DLLP performance seems to massively drop from Bag size 16 to Bag size 32, this looks a bit strange, what is the reason for this? Related to this, recently I have come across another LLP paper [1] (arxived after NeurIPS submission), which also implements the baseline DLLP for CIFAR 10 and CIFAR 100, the performance for the same baseline seems to be a lot better in that paper. Can the authors comment on the differences in the experimental setting (architecture, training epochs etc)? If the reviewers can clarify my comment and questions, I am happy and open to raising my score. [1] Deep multi-class learning from label proportions. Gabriel Dulac-Arnold, Neil Zeghidour, Marco Cuturi, Lucas Beyer, Jean-Philippe Vert Rebuttal: I have increased my score, as the authors have clarified my comments on motivation, and experiments.

Reviewer 3



The LLP setting is very relevant in practice, having many applications and although there exist good algorithms available the idea of using GANs to address the problem is new to me and makes good sense. The formulation seems to be technically sound and the adaptation of GAN loss functions for the LLP setting seems technically correct. The paper isn't very clearly written, and I suspect the rusty English may be partly responsible for that. But only partly since there are important practical considerations missing from the presentation. For instance, how are the bags created for the purpose of the experiment? I could not find an explanation. Having worked on this problem before, I'm familiar with the results showing how sensitive the LLP setting is to how similar the bag conditional distributions of the features are across different bags (e.g. ref [20] in the paper). For instance, if bags are generated randomly the results will be pretty bad because the information added by the label proportions of a bag won't add anything to those of other bags. It is crucial to know how the bags were generated in situations where the bags are artificially created, as in the experimental settings in this paper. This is however not discussed. That aside, the results comparing the proposed approach with a DLLP baseline, also using deep learning but without adversarial learning, suggest in general a significant improvement. Still, there is little insight as to why that may be the case. How much of this gap is an artefact of how the bags were chosen? Hard to say. Also, the experiments only cover datasets in which LLP isn't actually needed - all labels are known. This is useful to be able to evaluate the performance of the algorithms since labels are available, but apparently no effort is made to actually motivate the setting with any realistic scenario in which LLP is required. I think this is a problem. If the setting is worthy of being pursued, then why is it so hard to find a dataset in which there is a genuine reason to use the LLP setting? Overall I feel this is an ok paper but the quality of the presentation and the gaps in the experimental section make it not quite ready for neurips. Finally, I do not believe the answers to the reproducibility checklist are consistent with what is offered. For instance, how can I reconstruct the bags?

[Author Response · NeurIPS 2019]

We are grateful to the anonymous reviewers for the insightful feedback on our work. Please see our responses below.

**Question 1** What is the motivation of using GAN (adversarial learning) for learning from label proportions?

**Response** There are mainly three reasons for using GAN to solve LLP problem. Firstly, as described in paragraph 3 of
the Introduction, GAN is an elegant recipe for solving WeLL problems, especially semi-supervised learning [1]. From
this viewpoint, our approach is in line with the idea of applying GAN to incomplete label scenarios. More important, the
success of generative models for WeLL stems from the explicit or implicit representation learning, which has been an
essential method for unsupervised learning for a long time. In LLP-GAN, the conv layers in discriminator can perform
as a feature extractor for downstream tasks, which is proved to be efficient [2]. Hence, our work can be regarded as
**solving LLP based on representation learning with GAN**. In this scheme, generated fake samples encourage the
discriminator to not only detect the difference between the real and the fake instances but also distinguish *true K classes*
for real samples (*K+1 classifier*). Thirdly, most LLP methods assume that the bags are i.i.d., which cannot sufficiently
explore the underlying distribution in the data and may contradict in some applications. Instead, the generator in
LLP-GAN is designated to learn the data distribution through the adversarial scheme without this assumption.

**Question 2** What is the performance of the baseline of using entropy regularization for DLLP?

**Response** Firstly, this straightforward improvement for DLLP is a side contribution of our work. We consider not to
include it as a baseline because the experimental results suggest that the original DLLP has already converged to the
solution with fairly low instance-level entropy, which makes the regularization term redundant. Please see Figure 1.

**Question 3** What is the advantage of performing gradient method on the lower bound instead of original objective?

**Response** The most important advantage of this trick is it allows to *perform SGD upon every instance* instead of every
bag. It can greatly accelerate the convergence. However, in DLLP, we follow the original setting without this trick.

**Question 4** Please offer more details on the experimental results.

**Response** Two issues should be clarified for experiments. Firstly, as shown in Figure 3 of our paper, the results
demonstrate oscillation as bag size soaring. This phenomenon indicates a common drawback of deep models: For more
complex objective surfaces (more possible label candidates), normally the convergence will be dramatically getting
worse, due to more chances to attain local minima or saddle points of the objective. Secondly, because our results are
based on original datasets without data augmentation, the reported DLLP performance is worse than that in [3].

**Question 5** How much influence of the bag construction on the final results?

**Response** Indeed, the distribution of proportions has an huge impact on LLP algorithm performance. Hence, fixing bag
size, we randomly construct bags for multiple times and present the accuracy performance in Table 1. The result shows
the stability of our method. Currently, we can only artificially build LLP datasets from supervised ones. However, the
gap between the importance of LLP in real-life and lack of specific LLP datasets exactly suggests the meaning of our
work: It is worthy of devoting efforts to further study in order to draw more attention from the community.

| Bag Size (# of Random) | # of Errors | Accuracy (%) (Deviation) | Baseline (CNN) |
|---|---|---|---|
| 16 (7) | 106 | 98.94 (0.0285) | |
| 32 (22) | 124 | 98.76 (0.0542) | 99.64 |
| 64 (45) | 147 | 98.53 (0.11) | |
| 128 (85) | 335 | 96.65 (0.4) | |

Table 1: The performance on accuracy with deviation under multiple random bag generations on MNIST. (Due to the time limitation of response, # of random are differently chosen.)

Figure 1: Sum of instance-level entropy on MNIST.

## 33 References

[1] Tim Salimans, Ian Goodfellow, Wojciech Zaremba, et al. Improved techniques for training GANs. In *Advances in*
*Neural Information Processing Systems*, pages 2234–2242, 2016.

[2] Alec Radford, Luke Metz, and Soumith Chintala. Unsupervised representation learning with deep convolutional
generative adversarial networks. *arXiv preprint arXiv:1511.06434*, 2015.

[3] Gabriel Dulac-Arnold, Neil Zeghidour, Marco Cuturi, Lucas Beyer, and Jean-Philippe Vert. Deep multi-class
learning from label proportions. *arXiv preprint arXiv:1905.12909*, 2019.


[Meta-Review · NeurIPS 2019]

All three reviewers agreed that the idea of using adversarial learning for learning from label proportions is novel and interesting. For the final version, we would recommend the proposed method to be validated on: 1) structured and non-random bag assignment setting (e.g. Patrini et al., (Almost) no label no cry, NIPS 2014); 2) tabular data.